# Association of PM_2.5_ and Its Chemical Compositions with Metabolic Syndrome: A Nationwide Study in Middle-Aged and Older Chinese Adults

**DOI:** 10.3390/ijerph192214671

**Published:** 2022-11-08

**Authors:** Qian Guo, Yuchen Zhao, Tao Xue, Junfeng Zhang, Xiaoli Duan

**Affiliations:** 1School of Energy and Environmental Engineering, University of Science and Technology Beijing, Beijing 100083, China; 2Institute of Reproductive and Child Health/Ministry of Health Key Laboratory of Reproductive Health and Department of Epidemiology and Biostatistics, School of Public Health, Peking University, Beijing 100083, China; 3Nicholas School of the Environment and Global Health Institute, Duke University, Durham, NC 27708, USA; 4Duke Kunshan University, Kunshan 215316, China

**Keywords:** particulate matter, metabolic syndrome, adults, composition, cross-sectional

## Abstract

Studies on the association of PM_2.5_ and its compositions with metabolic syndrome (MetS) were limited, and it was unclear which was the most hazardous composition. In this study, we aimed to investigate the association between PM_2.5_ and its compositions with MetS and identified the most hazardous composition. In this study, we included 13,418 adults over 45 years across 446 communities from 150 counties of 28 provinces in nationwide China in 2015. MetS was defined based on the five indicators of the Joint Interim Societies, including: blood pressure (SBP (systolic blood pressure) and DBP (diastolic blood pressure)); fasting blood glucose (FBG); fasting triglyceride (FTG); high density lipoprotein cholesterol (HDL-C); and waist circumference (WC). We used chemical transport models to estimate the concentration of PM_2.5_ and its compositions, including black carbon, ammonium, nitrate, organic matter, and sulfate. We used a generalized linear regression model to examine the association of PM_2.5_ and its compositions with MetS. In this study, we observed that the average age was 61.40 (standard deviation (SD): 9.59). Each IQR (29.76 μg/m^3^) increase in PM_2.5_ was associated with a 1.27 (95% CI: 1.17, 1.37) increase in the odds for MetS. We indicated that black carbon showed stronger associations than other compositions. The higher associations were observed among women, participants aged less than 60 years, who lived in urban areas and in the Northeast, smokers, drinkers, and the obese populations. In conclusion, our findings identified the most harmful composition and sensitive populations and regions that required attention, which would be helpful for policymakers.

## 1. Introduction

Metabolic syndrome (MetS) is an indicator combined with increased blood pressure, high blood glucose and triglyceride, excess waist circumference (WC), and low levels of cholesterol [1]. The prevalence of MetS is increasing globally, estimated to be 20–25% in the adult population [2,3]. In China, there was 24.5% of the population suffering from MetS in 2016 [4].

MetS is a global public health issue, which may lead to cardiovascular disease, diabetes, stroke, asthma, cancer, and related mortality [5,6], leading to the heavy burden of disease. The prevalence of MetS also increased with age due to the commonalities in biochemical changes in aging process and MetS [7]. Therefore, it is urgent to identify the risk factors of MetS, especially among the aging population. Previous studies indicated that MetS would be attributable to genetic factors, unhealthy diet, unhealthy lifestyle (smoking, drinking, and sedentary), and inadequate sleep [8].

A growing number of epidemiological studies suggested air pollution was a risk factor for MetS [9,10,11]. Some studies showed air pollutants might induce MetS risk [12,13]. However, the evidence that investigated the associations of air pollution with MetS was still limited and inconsistent, especially among developed countries with higher air pollution. Up to now, only several studies have examined the associations of air pollution with MetS among Chinese adults [10,14,15].

The inconsistencies in results may be due to the difference in the study population, region, sample, period, and level of air pollution. Another explanation was that fine particulate matter (PM_2.5_) was a complex mixture including a series of components with various toxicities [16,17], which may lead to inconsistent results across different regions. However, most epidemiological studies only focused on the mass concentration of PM_2.5_ [9,10,11,18].

In recent years, some researchers have started to focus on the compositions of PM_2.5_ and explore the health outcomes attributed to various compositions, such as mortality, birth weight, and respiratory health [19,20,21]. To our knowledge, there were only a few studies that explored the association of PM_2.5_ compositions with indicators of MetS, such as blood pressure [22], blood glucose [23], and blood lipids [24]. However, there has been no study to explore the associations of PM_2.5_ compositions with MetS up to now. Hence, it is essential to examine the association between compositions of PM_2.5_ and MetS, and identify the most harmful composition, which would be helpful to policy-makers in carrying out the control of air pollution.

This study provided the first chance to explore the association of PM_2.5_ and its compositions with MetS among middle-aged and older adults based on the China Health and Retirement Longitudinal Study (CHARLS). We also aimed to identify the most hazardous components across regions and determine the most vulnerable region which should be focused on.

## 2. Materials and Methods

### 2.1. Study Participants

The dataset was taken from wave 3 of the China Health and Retirement Longitudinal Study (CHARLS) in 2015, which covered 150 counties/districts, 450 villages/urban communities across the country, involving 13,418 middle-aged and older adults above 45 years old in nationwide China by a multistage probability sampling strategy. The previous study described the details of the study population selection [25]. CHARLS collected the sociodemographic variables, health status, and household information using trained interviewers according to a standard protocol. The data of CHARLS can be obtained at http://opendata.pku.edu.cn publicly (accessed on 8 June 2022). All participants signed the written informed consent. The Ethical Review Committee of Peking University approved the study (IRB00001052-11015).

### 2.2. Estimation of PM_2.5_ and Its Composition

We estimated the concentration of PM_2.5_ and its compositions using aerosol optical depth (AOD) and conversion factors derived from the chemical transport models and evaluated against the ground observations [26]. In this study, the compositions included black carbon (BC), ammonium (NH_4_^+^), nitrate (NO_3_^-^), organic matter (OM), and sulfate (SO_4_^2−^). The data of PM_2.5_ and its composition were publicly available from the Tracking Air Pollution in China (TAP) (http://tapdata.org/, last access: 20 May 2022). The spatial resolution of PM_2.5_ and its composition were at 0.1° × 0.1° grid. The concentration of each composition was calculated as follows (Equation (1)):(1)$${\mathrm{Composition}}_{\mathrm{satellite}}^{k}={\mathrm{AOD}}_{\mathrm{satellite}}\times \frac{{\mathrm{Composition}}_{\mathrm{CTM}}^{k}}{{\mathrm{AOD}}_{\mathrm{CTM}}}$$

In Equation (1), “satellite” and “CTM” refer to the data from the satellite and model, respectively; k refers to the different compositions of PM_2.5_; the conversion factors between PM_2.5_ compositions and AOD were estimated from the emission inventory over China [27].

In this study, the concentration of air pollutants was available at the city level because of confidentiality. We estimated the concentrations of PM_2.5_ and its compositions according to the spatiotemporal coordinates and the date of the interview on the monthly average. We calculated the average concentration of PM_2.5_ and its composition during the 1-year preceding the survey month as a surrogate for the long-term exposure concentration (Lag 1). We also simultaneously estimated the annual concentrations of PM_2.5_ and its components at different exposure windows (from Lag 2 to Lag 5).

### 2.3. Definition of MetS

Metabolic syndrome (MetS) was jointly defined by five indicators, including blood pressure (SBP (systolic blood pressure) and DBP (diastolic blood pressure)); fasting blood glucose (FBG); fasting triglyceride (FTG); high density lipoprotein cholesterol (HDL-C); and waist circumference (WC). We determined that participants had MetS if they presented with three or more of the abovementioned factors. In this study, we chose the definition of MetS according to the joint definitions from different organizations and used the cut-off points of WC from China [28]. Specifically, (1) SBP ≥ 130 mmHg or DBP ≥ 85 mmHg; (2) WC > 85 cm for men and 80 cm for women, respectively; (3) FBG > 100 mg/dL; (4) FTG > 150 mg/dL; (5) HDL-C < 40 mg/dL for men and 50 mg/dL for women, respectively. Different organizations define MetS differently, and the detailed criteria for these definitions are shown in Appendix A.

### 2.4. Covariates

Potential confounders were collected via a questionnaire by well-trained technicians. We obtained covariates including socioeconomic variables sex, age, urbanicity, marital status, educational level, the status of drinking and smoking, and the fuel type of household cooking/heating. The specific classification of each variable was presented in previous studies [29]. The fuel was classified into clean fuel (marsh gas, solar energy, natural gas, natural gas, liquefied petroleum gas, electricity, or municipal heating) and solid fuel (crop residue, coal, wood, or solid charcoal).

### 2.5. Statistical Analysis

The generalized linear model was used to explore the association of PM_2.5_ and its compositions with MetS adjusted for age, sex, urbanicity, educational level, marriage status, smoking, drinking, fuel type of cooking and heating, and physical activity. We described the results as odds ratios (OR) of MetS per IQR increment of PM_2.5_ and its compositions. We also conducted region-stratified analyses. We divided the whole of China into four districts: East; Middle; West; and Northeast.

We conducted a series of sensitivity analyses: (1) First, we investigated the association at different exposure windows (Lag 2, Lag 3, Lag 4, and Lag 5); (2) Second, we developed the analyses in different model-adjusted confounders; (3) We conducted the stratified analyses by sex, age group, urbanicity, smoking status, drinking status, and obese or not; (4) We also repeated the analyses to explore the association using the different definition of MetS from other organizations. All data processes were performed using R 4.1.1 (R Foundation for Statistical Computing, Vienna, Austria). The two-tailed *p* value < 0.05 was seen as statistical significance.

## 3. Results

### 3.1. The Basic Description

The characteristics of participants are depicted in Table 1. There were 4044 (30.14%) participants who had MetS in the population of 13,418. Participants were 61.40 (standard deviation (SD): 9.59) years old. We compared the differences in characteristics between participants with or without MetS and found a significant difference between the two groups. We also observed that the participants with MetS had a higher body mass index (BMI), undertook less physical activity, were more likely to be women, living in a rural area, married, and to be from the population with a lower educational level.

Table 2 shows the description of PM_2.5_ and its compositions. The mean ± SD concentration was 49.31 ± 19.63 μg/m^3^, 2.36 ± 0.70 μg/m^3^, 7.45 ± 3.11 μg/m^3^, 10.74 ± 5.11 μg/m^3^, 11.93 ± 4.23 μg/m^3^, 9.24 ± 3.34 μg/m^3^ for PM_2.5_, BC, NH_4_^+^, NO_3_^-^, OM, and SO_4_^2-^, respectively. The concentration of PM_2.5_ and its compositions at different exposure windows did not differ significantly (Appendix A). Appendix A shows the concentrations of PM_2.5_ and its compositions in different regions. Most compositions had the highest concentration in the East, while the lowest concentration was observed in the West (Appendix A). We also observed that the concentrations of all constituents in the East and middle regions exceeded the national average, while the concentrations in the West and Northeast regions were lower than the national average.

### 3.2. Association between PM_2.5_ and Its Compositions on MetS

Our study indicated significant positive associations of PM_2.5_ and its compositions with the risk of MetS (Figure 1). The adjusted OR of MetS was 1.27 (95% CI: 1.17, 1.37) with an IQR (29.76 μg/m^3^) increment in PM_2.5_. In this study, we observed that BC was the most harmful composition because BC showed the strongest associations among all compositions. In the main model, per IQR (1.06 μg/m^3^) the increment in BC was associated with a 1.25 (95% CI: 1.16, 1.36) increase in the risk of MetS. To identify the most vulnerable regions, we conducted region-stratified analyses and found that associations varied significantly in different regions. We also observed that PM_2.5_ and all the compositions showed the greatest effects on MetS in the Northeast region, while the weakest effects were observed in the Western region. Take BC as an example, in the fully adjusted models, an IQR (1.06 μg/m^3^) increase in BC was associated with a 65% (95% CI: 1.39, 1.96) and 19% (95% CI: 1.06, 1.34) increase in the risk of MetS in the Northeast and Western region, respectively.

We also conducted a set of sensitivity analyses. We explored the association of PM_2.5_ and its compositions with MetS by different models after adjustments for various covariates (Appendix A). We found that PM_2.5_ and its compositions retained significant positive associations across different models. We also repeated the analyses using the concentration at different exposure windows, and observed that the positive associations of PM_2.5_ and its chemical compositions with MetS remained stable (Appendix A). Finally, the repeated analyses using the different definition of MetS also showed the positive associations (Appendix A). The above-mentioned series of sensitivity analyses indicated the robustness of our findings.

### 3.3. Stratified Analyses by Subgroups

Stratified analyses were performed on sex, age group, urbanicity, smoking status, drinking status, and obese or not (Figure 2). Although there were no significant differences in different subgroups, we still observed a consistent trend. It was noted that a greater association of PM_2.5_ and its compositions with MetS was observed among women, participants aged less than 60 years, who lived in urban areas, were smokers, drinkers, and obese.

## 4. Discussion

In this large nationwide epidemiology study, we observed PM_2.5_ and its chemical composition associated with the increase in the risk of MetS for the first time. BC showed the strongest associations. The population living in the Northeast region showed the greatest effects. Generally, stronger associations were observed in women, participants aged less than 60 years, who lived in urban areas, were smokers, drinkers, and obese. This study provided further evidence on the association between PM_2.5_ and MetS, and filled the gap on the topic of the association between PM_2.5_ compositions and MetS. This study provided the first chance to identify the most harmful composition, vulnerable region, and population that required attention, which would help influence policy-making decisions.

Some studies have investigated the association of PM_2.5_ with MetS, which was in line with our findings. The Normative Aging Study conducted in Boston found that each 1-μg/m^3^ increment in PM_2.5_ increased by 27% (95% CI: 6%, 52%) the risk of MetS. Another study in Korea indicated that each 10-μg/m^3^ increment in PM_2.5_ increased by 7% the risk of MetS [30]. However, most existing studies were found among developed countries, and limited studies were performed in the highly polluted areas. Up to now, only several studies were conducted among Chinese adults [10,14,15]. All the studies conducted in China found that PM_2.5_ increased the risk of MetS, which was in line with our findings.

In this study, BC showed the strongest associations of PM_2.5_ and its chemical compositions with MetS. BC was mainly derived from the incomplete burning of fossil fuels and biomass, and may play the most hazardous effect due to its tiny size [31,32]. Moreover, incomplete combustion may generate organic pollutants with greater toxicity carried on BC [33]. For other compositions, we also found a significant positive association with MetS. The OM was primarily from burning unclean fuel (biomass or coal), which may induce lipid metabolic disturbances [34,35]. The acidification and oxidation processes of sulfate dioxide (SO_2_) or nitrogen oxides (NO_x_) may generate inorganic water ions, such as NH_4_^+^, SO_4_^2−^, and NO_3_^-^ [36,37]. Lipid metabolism was a possible mechanism by which the inorganic water ions affect MetS [38,39].

It is essential to identify the vulnerable population attributed to air pollution. In the stratified analyses, we observed stronger associations among women, subjects aged less than 60 years, who lived in urban areas, who were smokers, drinkers, and obese. We observed a greater association in women, which could be attributed to their smaller airways [40]. Prior evidence had reported greater associations in the elderly. However, we observed stronger associations among the younger population, consistent with some other studies [10,41]. A possible explanation is that the older population have a reduced response to nervous system stimuli, which may be more resistant than the younger population [10,42]. The participants who lived in urban areas were more sensitive and showed greater associations due to higher level of air pollution [43]. Our study also found that smokers, drinkers, and the obese population may have greater associations, which suggested unhealthy lifestyles may aggravate the harmful effects of air pollutants. The potential explanation is that unhealthy lifestyles can induce insulin resistance, leading to systemic inflammation and oxidative stress [44,45]. This study found the greatest associations of PM_2.5_ and all the compositions on MetS in the Northeast region. The level of PM_2.5_ and its compositions were higher in the northeast area due to more emissions from the burning of fossil fuels [46,47]. Meanwhile, the Western region showed the weakest associations due to the low population density and low level of air pollution [48].

Although the underlining mechanisms between air pollution and MetS were still unclear, some studies suggested several potential biological pathways. First, air pollutants would enter the human body and induce oxidative stress and inflammation, which would contribute to developing MetS [49,50]. Second, air pollution might disrupt insulin signaling and cause insulin resistance via inducing the production of endogenous pro-inflammatory mediators and vasoactive molecules [51,52]. Third, air pollution might activate the pathways of development of MetS by interfering with DNA methylation levels [12,53].

The present study has notable strengths. First, this is the first epidemiological study to investigate the association of PM_2.5_ chemical compositions with MetS among adults. This study also provided new evidence on the association between PM_2.5_ and MetS based on the nationwide sample. Second, this study collected a set of potential confounders using the standard questionnaire. Third, this study covered national spatial data and included a large sample, which provided a chance to obtain the results across the nation.

This study has several limitations that require attention. First, the characteristics of the cross-sectional study design did not allow us to obtain causal inferences. Second, considering participants’ confidentiality, the geographic information was only available at the city level, which may induce the misclassification of exposure. Third, although we have investigated many confounders, there were still unmeasured factors, including genes and other gaseous pollutants. Last, we did not include the gaseous components of PM_2.5_ because of the lack of data. Further studies are needed to confirm our findings and add more evidence on the association of PM_2.5_ and its compositions with MetS.

## 5. Conclusions

In this national sample of Chinese adults above 45 years old, 30.14% had MetS. This is the first study that observed that the PM_2.5_ chemical compositions increased the risk of MetS among Chinese adults. We also found that BC showed the most hazard effects on the risk of MetS, which revealed that reducing BC emissions is an effective measure to reduce the risk of obesity. Women, participants aged less than 60 years, who lived in an urban area and the Northeast region, smokers, drinkers, and the obese population were more susceptible to PM_2.5_ and its compositions, and needed more attention. More epidemiological and toxicological studies were required to confirm our findings further and add new evidence.

## Figures and Tables

**Figure 1 ijerph-19-14671-f001:**
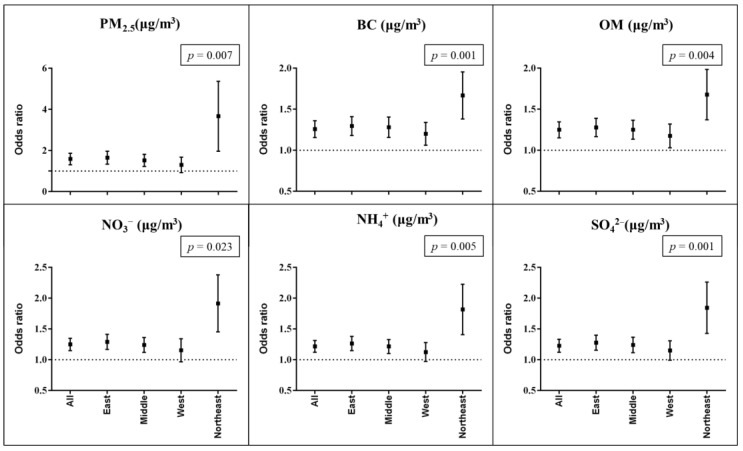
The association of metabolic syndrome (MetS) with an IQR incremental change in 1-year average PM_2.5_ and its constitution, stratified by different regions. The bars show main effect estimates and 95% confidence intervals. The model was adjusted for age, sex, urbanicity, educational level, marriage status, smoking, drinking, cooking fuel type, heating fuel type, and physical activity. Abbreviations: BC: Black Carbon; NH_4_^+^: Ammonium; NO_3_^−^: Nitrate; OM: organic matter; SO_4_^2−^: Sulfate. *p* value refers to the difference of the estimations between different regions.

**Figure 2 ijerph-19-14671-f002:**
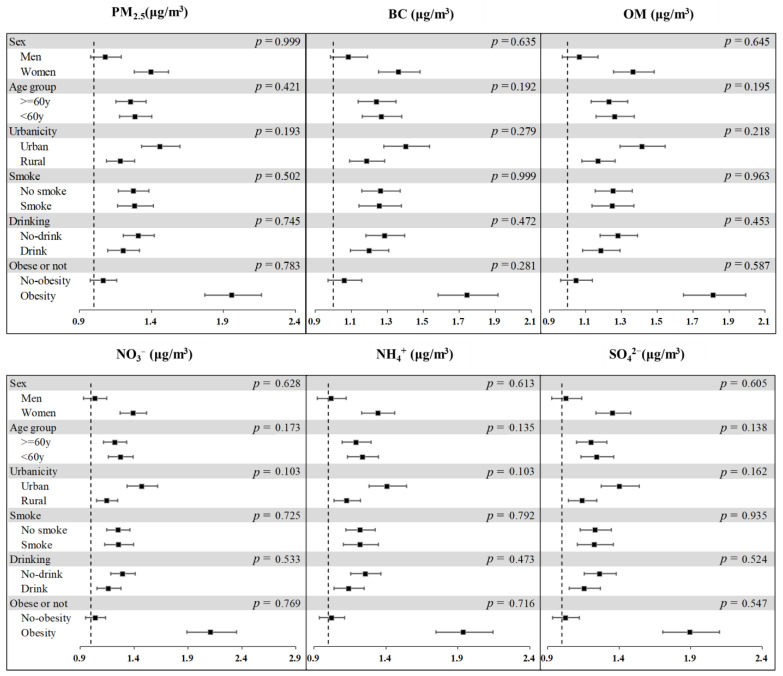
Odds ratios of metabolic syndrome (MetS) associated with an IQR incremental change in 1-year average PM_2.5_ and its constitution, classified by different subgroups. The bars show main effect estimates and 95% confidence intervals. The model was the fully adjusted model adjusted for age, sex, urbanicity, educational level, marriage status, smoking, drinking, cooking fuel type, heating fuel type, and physical activity. Abbreviations: BC: Black Carbon; NH_4_^+^: Ammonium; NO_3_^−^: Nitrate; OM: organic matter; SO_4_^2−^: Sulfate. *p* value refers to the difference of the estimations between different subgroups.

**Table 1 ijerph-19-14671-t001:** Characteristic of the participants.

	Non-MetS (*n* = 9374)	MetS (*n* = 4044)	*p* Value
Age, mean (SD), year	61.6 (9.79)	60.8 (9.07)	<0.001
Height, mean (SD), cm	158 (8.35)	158 (8.56)	0.392
Weight, mean (SD), kg	57.5 (10.5)	65.8 (11.3)	<0.001
Physical activity amount, mean (SD), MET min/week	119 (113)	95.5 (101)	<0.001
Sex (No. (%))			<0.001
Men	4681 (49.9%)	1492 (36.9%)	
Women	4691 (50.0%)	2552 (63.1%)	
Missing	2 (0.0%)	0 (0%)	
Urban or rural (No. (%))			<0.001
Urban	3227 (34.4%)	1869 (46.2%)	
Rural	6147 (65.6%)	2175 (53.8%)	
Marriage or not (No. (%))			0.28
Single	1217 (13.0%)	497 (12.3%)	
Married	8155 (87.0%)	3547 (87.7%)	
Missing	2 (0.0%)	0 (0%)	
Educational level (No. (%))			0.002
Primary or below	3952 (42.2%)	1609 (39.8%)	
Middle school	1721 (18.4%)	815 (20.2%)	
High school	705 (7.5%)	348 (8.6%)	
Collage or above	134 (1.4%)	72 (1.8%)	
Missing	2862 (30.5%)	1200 (29.7%)	
Heating fuel ^1^ (No. (%))			<0.001
Solid	5445 (58.1%)	2139 (52.9%)	
Clean	1516 (16.2%)	732 (18.1%)	
Missing	2413 (25.7%)	1173 (29.0%)	
Cooking fuel ^1^ (No. (%))			<0.001
Solid	4277 (45.6%)	1514 (37.4%)	
Clean	5081 (54.2%)	2520 (62.3%)	
Missing	16 (0.2%)	10 (0.2%)	
Smoke or not (No. (%))			<0.001
Smoke	2635 (28.1%)	767 (19.0%)	
No-smoke	5336 (56.9%)	2708 (67.0%)	
Missing	1403 (15.0%)	569 (14.1%)	
Drinking or not (No. (%))			<0.001
Drink	3480 (37.1%)	1184 (29.3%)	
Ever-drink	1015 (10.8%)	442 (10.9%)	
No-drink	4860 (51.8%)	2411 (59.6%)	
Missing	19 (0.2%)	7 (0.2%)	

Abbreviations: MetS, metabolic syndrome; SD, standard deviation. Notes: ^1^ Cooking fuel was classified into clean fuel (marsh gas, natural gas, liquefied petroleum gas, or electricity) and solid fuel (crop residue, coal, wood, or solid charcoal). Heating fuel was categorized as solid fuel (coal, crop residue, wood, or solid charcoal) and clean fuel (natural gas, solar energy, liquefied petroleum gas, electric, or municipal heating).

**Table 2 ijerph-19-14671-t002:** The 1-year average concentration of the PM2.5 and its chemical constituent.

Air Pollutants	Mean	SD	Min	P_5_	P_25_	Median	P_75_	P_95_	Max
PM_2.5_	49.31	19.63	19.84	22.14	32.25	45.87	61.89	84.98	93.27
BC	2.36	0.70	0.98	1.37	1.82	2.21	2.88	3.63	3.99
NH_4_^+^	7.45	3.11	2.76	3.04	4.83	7.43	9.57	12.92	14.03
NO_3_^−^	10.74	5.11	3.07	3.56	6.27	10.28	14.26	19.56	21.38
OM	11.93	4.23	4.95	6.00	8.56	11.46	14.77	20.12	21.70
SO_4_^2−^	9.24	3.34	3.07	4.78	6.35	8.64	11.77	15.27	16.61

Abbreviations: BC: Black Carbon; NH_4_^+^: Ammonium; NO_3_^−^: Nitrate; OM: organic matter; SO_4_^2−^: Sulfate.

## Data Availability

Not applicable.

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
