# Peer review of "Association of PM2.5 and Its Chemical Compositions with Metabolic Syndrome: A Nationwide Study in Middle-Aged and Older Chinese Adults"

_ijerph, 2022, doi:10.3390/ijerph192214671_

Round 1

Reviewer 1 Report

Dear Atuhors,

The paper brings an exploration between metabolic syndrome and chemical compositions, and has an interesting scientific issue. For literature is an important contribution because the relevance in consider several factors from population presented on the paper.
However, the paper fell deeply into a mistake in not letting clear the methodological draw. The authors only considered 13.418 adults and a lack of other information as period in collect, details about protocol interviewers, cities included on surveys are not clear in section 2.1.
In relation to the literature review, I feel necessary rise similar studies to include in this research.
The presentation of Results emphasizes the tables and figures, but the description of results is poor and the real contribution doesn't look clear.

Author Response

The paper brings an exploration between metabolic syndrome and chemical compositions, and has an interesting scientific issue. For literature is an important contribution because the relevance in consider several factors from population presented on the paper.

Response:Thank you for your valuable and positive comments. We appreciated the detailed and valuable comments and suggestions, which are very helpful for improving the quality of this manuscript. We have carefully reviewed all the questions and suggestions and made corresponding corrections or changes.

Q1: However, the paper fell deeply into a mistake in not letting clear the methodological draw. The authors only considered 13.418 adults and a lack of other information as period in collect, details about protocol interviewers, cities included on surveys are not clear in section 2.1.

Response:Thank you very much for your comments. We feel so sorry that we didn’t describe the detail information of the method. To make it clear, we have rewritten the section 2.1 (Line 79-83): “The dataset was from wave 3 of the China Health and Retirement Longitudinal Study (CHARLS) in 2015, which covered 150 countries/districts, 450 villages/urban communities across the country, involving 13,418 middle-aged and older adults above 45 years old in nationwide China by multistage probability sampling strategy. The previous study described the details of the study population selection [25].”

Q2: In relation to the literature review, I feel necessary rise similar studies to include in this research.

Response:Thank you very much for your comments. As far as we know, there was no study have explored the association of PM2.5 and its compositions with MetS. However, there was an increasing number of epidemiological studies have begun to explore the association between PM2.5 chemical constituents and a range of health outcomes, particularly those affecting lung function, mortality, and birth weight. However, the existing studies are inconsistent, and the harmful health effects of various PM2.5, constituents remain unclear. We also added this in the introduction sections (Line 63-71): “In recent years, some researchers have started to focus on the compositions of PM2.5 and explore the health outcomes attributed to various compositions, such as mortality, birth weight, and respiratory health [19-21]. To our acknowledge, there were a few studies have explored the association of PM2.5 compositions on indicators of MetS, such as blood pressure [22], blood glucose [23], and blood lipids [24]. However, there has been no study to explore the associations of PM2.5 compositions on MetS up to now. Hence, it is essential to examine the association between compositions of PM2.5 and MetS, and identify the most harmful composition, which would be helpful to policy-maker to carry out the control of air pollution.”

       In the discussion, we also mentioned similar results in the paragraph 2 and 3 to compare with our findings.

Q3: The presentation of Results emphasizes the tables and figures, but the description of results is poor and the real contribution doesn't look clear.

Response:Thank you very much for your comments. To make it clearer, we have rewritten the results section as follows:

In line 149-151: “We have compared the differences in characteristics between participants with or without MetS and found a significant difference between the two groups.”

Line 166-169: “We also observed that the concentrations of all constituents in the East and middle regions exceeded the national average, while the concentrations in the west and northeast regions lower than the national average.”

Line 175-185: “In this study, we observed that BC was the most harmful composition because BC showed the strongest associations among all compositions. In the main model, per IQR (1.06μg/m3) increment in BC was associated with a 1.25 (95% CI: 1.16,1.36) increase in the risk of MetS. To identify the most vulnerable regions, we conducted region-stratified analyses and found that associations varied significantly in different regions. We also observed that PM2.5 and all the compositions showed the greatest effects on MetS in the Northeast region, while the weakest effects were observed in the western region. Take BC as an example, in the fully adjusted models, an IQR (1.06μg/m3) increase in BC was associated with a 65% (95% CI: 1.39: 1.96) and 19% (95% CI: 1.06: 1.34) increase in the risk of MetS in northeast and west region, respectively.”

Line 192-200: “We also conducted a set of sensitivity analyses. We explored the association of PM2.5 and its compositions with MetS by different models after adjustments for various covariates (Table A.4). We found that PM2.5 and its compositions remained significant positive associations across different models. We also repeated the analyses using the concentration at different exposure windows, and observed that the positive associations of PM2.5 and its chemical compositions with MetS remained stable (Table A.5). Finally, the repeated analyses using the different definition of MetS also showed the positive associations (Figure A.1). The above mentioned a series of sensitivity analyses indicated the robustness of our findings.”

Reviewer 2 Report

The paper are interesting for large  number of target groups in a time of aggressive polution, the journal and title are suitable and the abstract content is well selected . The paper has other strong aspects related to well selected References Chapter and being as the authors afirmed the first epidemiological study  to investigate association of PM2.5 chemical compositions with MetS among adults. 

a)taking into considerations that the present  paper form is a too simple statistics introduce  more factors of association  influence   I do suggest gaseous components of PM2.5 as a function of industrial development  .

b) paper organization could be improved .A part of discussion has a better place in introduction being more suitable as the state of the art of the subject

c) English language and style could be polished 

Author Response

The paper are interesting for large number of target groups in a time of aggressive pollution, the journal and title are suitable and the abstract content is well selected. The paper has other strong aspects related to well selected References Chapter and being as the authors affirmed the first epidemiological study to investigate association of PM2.5 chemical compositions with MetS among adults. 

Response:

Thank you for your positive comments and suggestions. We believed that based on your detailed comments, the manuscript could be improved a lot than before. We have already revised all of them as you suggested.

Q1: taking into considerations that the present  paper form is a too simple statistics introduce  more factors of association  influence   I do suggest gaseous components of PM2.5 as a function of industrial development.

Response:Thank you very much for your comments. We agreed with you that the gaseous components of PM2.5 was also important. However, in this paper, we obtained the concentration of PM2.5 and its compositions from the TAP dataset, which only included the black carbon (BC), ammonium (NH4+), nitrate (NO3-), organic matter (OM), and sulfate (SO42-). We didn’t have the data of the gaseous components of PM2.5. We also added this in the limitation (Line 280-281): “Last, we didn’t include the gaseous components of PM2.5 because of the lack of data.”

There were many studies conducted the association between above mentioned compositions on other health outcomes, such as obesity, lung function and mortality. However, there was no study explored the association of PM2.5 and its compositions with MetS. Although we didn’t use the complex statistics method in this study, this is the first study to explore the association PM2.5 and its compositions with MetS, which made contribution on identifying the most harmful composition and sensitive population and region that needed attention.

We agreed with your comments that more studies should be conducted to explore the association between gaseous components of PM2.5 and MetS in the further, which could provide new evidence on this topic.

Q2: paper organization could be improved .A part of discussion has a better place in introduction being more suitable as the state of the art of the subject.

Response:Thank you very much for your comments. We agreed with your comments. To make it clear, we have re-organize the paper. We put the paragraph “In recent years, some researchers have started to focus on the compositions of PM2.5 and explore the health outcomes attributed to various compositions, such as mortality, birth weight, and respiratory health [19-21]. To our acknowledge, there were a few studies have explored the association of PM2.5 compositions on indicators of MetS, such as blood pressure [22], blood glucose [23], and blood lipids [24]. However, there has been no study to explore the associations of PM2.5 compositions on MetS up to now. Hence, it is essential to examine the association between compositions of PM2.5 and MetS, and identify the most harmful composition, which would be helpful to policy-maker to carry out the control of air pollution.” in the introduction section to make it more suitable.

Q3: English language and style could be polished.

Response:Thank you very much for your comments. To improve the English language of this paper, we have already invited the professional experts in this field help us to revised the whole manuscript.

Reviewer 3 Report

This paper explored the association of PM2.5 and its chemical compositions with MetS. This is an interesting study and its quality is good. However, I would like to see the following improvements before making a concrete decision on the manuscript.

1.    Abstract should be restructured so as to provide a summary of the objectives, findings and tests. The authors use sentences in Lines 14-21 to introduce the research design, but with so much ink, it still does not clearly show how authors carry out this research.

2.    Some statements are inappropriate. There are too many long sentences in this paper, which are not clear enough. Authors need to sort out the long sentences in the paper. I would encourage the authors to have the language checked by a native speaker.

3.    What is the reason for choosing the data of CHARLS in 2015 for research? After browsing the official website of CHARLS, it provides the latest CHARLS data up to 2018. Why not use the CHARLS data in 2018?

4.    Section 2.5 and the first paragraph of Section 3.1 both provide descriptive analysis of Table 1. Can they be combined?

5.    At present, the conclusions of this paper are limited due to the choice of research methods. The authors may consider an econometric approach to identify causal effects of PM2.5 and its components on the prevalence of MetS.

Author Response

This paper explored the association of PM2.5 and its chemical compositions with MetS. This is an interesting study and its quality is good. However, I would like to see the following improvements before making a concrete decision on the manuscript.

Response: Thank you for your positive comments and suggestions. We have checked all your detail comments and suggestions and revised all of them. We believe this manuscript improved a lot than before with your help.

Q1: Abstract should be restructured so as to provide a summary of the objectives, findings and tests. The authors use sentences in Lines 14-21 to introduce the research design, but with so much ink, it still does not clearly show how authors carry out this research.

Response: Thank you very much for your comments. To make it clear, we have rewritten the abstract as follows: “Studies on the associations of PM2.5 with metabolic syndrome (MetS) were limited. Meanwhile, no study explored the association of PM2.5 compositions with MetS, and it’s unclear which was the most hazardous composition. In this study, we aimed to investigate the association between PM2.5 and its compositions with MetS and provide new evidence. In this study, we included 13,418 adults over 45 years across 446 communities from 150 counties of 28 provinces in nationwide China in 2015 using the multistage probability sampling method. We collected the sociodemographic variables, health status, household information, and blood samples of subjects. We also measured the blood biomarkers. MetS was defined based on the Joint Interim Societies based on the five indicators, including Blood pressure [SBP (systolic blood pressure) and DBP (diastolic blood pressure)]; fasting blood glucose (FBG); fasting triglyceride (FTG); high density lipoprotein cholesterol (HDL-C) and waist circumference (WC). We also estimated the concentration of PM2.5 and its compositions using chemical transport models. The compositions included black carbon, ammonium, nitrate, organic matter, and sulfate. We used a generalized linear regression model to examine the association of PM2.5 and its compositions with MetS. In this study, we observed that the prevalence of MetS among aging population was 30.14%. The average age was 61.40 (standard deviation (SD): 9.59). Each IQR (29.76μg/m3) increase in PM2.5 was associated with a 1.27 (95% CI:1.17, 1.37) increase in odds for MetS. We indicated that black carbon showed stronger associations than other compositions. The population who lived in the Northeast was most sensitive to PM2.5 and its compositions. The greater associations were observed among women, participants aged less than 60 years, who lived in urban, smokers, drinkers, and obese populations. In conclusion, our findings identified the most harmful composition and sensitive population and region that needed attention, which would be helpful for policymaker.”

Q2: Some statements are inappropriate. There are too many long sentences in this paper, which are not clear enough. Authors need to sort out the long sentences in the paper. I would encourage the authors to have the language checked by a native speaker.

Response:Thank you very much for your comments. To improve the English language of this paper and make it clearer, we have already invited the professional experts in this field help us to revised the whole manuscript.

Q3: What is the reason for choosing the data of CHARLS in 2015 for research? After browsing the official website of CHARLS, it provides the latest CHARLS data up to 2018. Why not use the CHARLS data in 2018?

Response: Thank you for your comments. As you said, the CHARLS have already finished the survey in 2018 and released the data on the demographic information, healthy status and economical level. However, there was no data on the blood biomarkers. In this paper, we focused on the MetS, which was defined by Blood pressure [SBP (systolic blood pressure) and DBP (diastolic blood pressure)]; fasting blood glucose (FBG); fasting triglyceride (FTG); high density lipoprotein cho-lesterol (HDL-C) and waist circumference (WC). The FBG, FTG, and HDL-C was measured in blood sample. In the CHARLS data in 2018, there was no data on the blood so that we can’t define MetS. This is the reason why we choose the data in 2015 to conduct our analyses.

Q4: Section 2.5 and the first paragraph of Section 3.1 both provide descriptive analysis of Table 1. Can they be combined?

Response: Thank you for your comments. As you suggested, to make it clear, we have combined the it and rewritten the paragraph of section 3.1 in Line 149-153: “We have compared the differences in characteristics between participants with or without MetS and found a significant difference between the two groups. We also observed that participants with MetS had higher Body mass index (BMI), less physical activity, women, who living in rural, married, and populations with lower educational level.”

Q5: At present, the conclusions of this paper are limited due to the choice of research methods. The authors may consider an econometric approach to identify causal effects of PM2.5 and its components on the prevalence of MetS.

Response: Thank you very much for your comments. We agreed with you that we cannot identify the causal effects of PM2.5 and its components on the prevalence of MetS. Based on the cross-sectional study design, we only could explore the association between PM2.5 and its components on the prevalence of MetS. In the future, more cohort studies were needed to explore the causal effects between them.

Round 2

Reviewer 1 Report

Dear authors,

I finished my considerations related the paper.

Best regards

Author Response

Response:  Thank you for your comments and suggestions, so that the manuscript could be improved a lot. 

Reviewer 2 Report

The authors  improved significantly  their manuscipt presenting  a more complete abstract and a better organised way of their  data.  The study participants was modified introducing  more information and mentioning how the  ethical review committee of Peking University approved the study. English language was polished as well.   my recommendation nowadays is to be published as it is  

Author Response

Response:  Thank you for your valuable comments and suggestions, so that the manuscript could be improved a lot and could be published!

Reviewer 3 Report

In general, authors have addressed most of my concerns. It would be better if the Abstract could be more concise and the Conclusions more substantial.

Author Response

Response:  Thank you very much for your comments. To improve the manuscript, we have rewritten the abstract section and conclusion section.

For abstract, the new version was as follows: “Studies on the associations of PM2.5 and its compositions with metabolic syndrome (MetS) were limited, and it’s unclear which was the most hazardous composition. In this study, we aimed to investigate the association between PM2.5 and its compositions with MetS and identified the most hazardous composition. In this study, we included 13,418 adults over 45 years across 446 communities from 150 counties of 28 provinces in nationwide China in 2015. MetS was defined based on the Joint Interim Societies based on the five indicators, including Blood pressure [SBP (systolic blood pressure) and DBP (diastolic blood pressure)]; fasting blood glucose (FBG); fasting triglyceride (FTG); high density lipoprotein cholesterol (HDL-C) and waist circumference (WC). We used chemical transport models to estimate the concentration of PM2.5 and its compositions, including black carbon, ammonium, nitrate, organic matter, and sulfate. We used a generalized linear regression model to examine the association of PM2.5 and its compositions with MetS. In this study, we observed that the average age was 61.40 (standard deviation (SD): 9.59). Each IQR (29.76μg/m3) increase in PM2.5 was associated with a 1.27 (95% CI:1.17, 1.37) increase in odds for MetS. We indicated that black carbon showed stronger associations than other compositions. The greater associations were observed among women, participants aged less than 60 years, who lived in urban and in the Northeast, smokers, drinkers, and obese populations. In conclusion, our findings identified the most harmful composition and sensitive population and region that needed attention, which would be helpful for policymaker.”

For conclusion, the new version was as follows: “In this national sample of Chinese adults above 45 years old, 30.14% had MetS. This is the first study that observed that PM2.5 chemical compositions increased the risk of MetS among Chinese adults. We also found that BC showed the most hazard effects on risk of MetS, which revealed that reducing BC emissions is an effective measure to reduce the risk of obesity. Women, participants aged less than 60 years, lived in urban and Northeast region, smokers, drinkers, and obese populations were more susceptible to PM2.5 and its compositions, and needed more attention. More epidemiological and toxicological studies were required to confirm our findings further and add new evidence.”